# Leadership and Service Delivery in Times of Change

Sulaiman Olusegun Atiku [1,2,*], Collen Mulife Kurana [1] and Idris Olayiwola Ganiyu [2,3]

1   Harold Pupkewitz Graduate School of Business, Namibia University of Science and Technology, Windhoek 13388, Namibia
2   Department of Economic and Business Sciences, Walter Sisulu University, Mthatha 5100, South Africa
3   York Business School, York St John University, Lord Mayor's Walk, York YO31 7EX, UK
*   Correspondence: olusegunatiku@gmail.com or satiku@nust.na; Tel.: +264-612072332

**Abstract:** There has been a growing concern for excellent service delivery in the public sector. The challenges that are hindering service delivery in Town Councils have been attributed to political interference in appointments and ineffective leadership. The residents in the Town Council have been experiencing power outages and water shortages. The livelihoods of the residents and business operations have been hindered by these difficulties. This study investigates the influence of leadership on service delivery in a Town Council in Namibia. A qualitative research approach was adopted to analyse the influence of leadership practices on service delivery in times of change. A total of ten participants were sampled from a population of 117 staff members using a purposive sampling. Face-to-face interviews were conducted using a semi-structured interview guide. Participants views were tape recorded, transcribed into Microsoft Word, and analysed using thematic analysis via NVivo 12. The results showed that logistical difficulties, political interference, lack of human resources, and financial constraints, are the challenges hampering service delivery in the Town Council. The need for managers and supervisors to inspire, coach, mentor, and motivate their subordinates to enhance service delivery using a transformational leadership style is evident in the results. Therefore, the Town Council should prioritise budgeting, foster public-private partnerships, promote innovation, and ensure stakeholder engagement to enhance service delivery in the Town Council.

**Keywords:** leadership practices; Namibia; public sector; service delivery; stakeholders' engagement; Town Council; transformational leadership

## 1. Introduction

The ineffectiveness of leadership in Africa is a major issue that has led to a variety of problems in the region. Weak leadership results in poor governance in Africa, which is often characterised by weak institutions, weak rule of law, corruption, nepotism, lack of accountability, lack of transparency, and poor service delivery. Empirical evidence has linked poor infrastructural development to poor leadership, corruption, and unsustainable infrastructure projects, resulting in poor service delivery (Mbandlwa 2020). However, Folarin (2010) perceived leadership as an intervening variable as it reduces or enhances service delivery depending on the policy, decision, and the implementation. Service delivery, as used in this context, refers to the provision of basic social services such as electricity, water, and other infrastructural facilities that are provided by the government.

Namibia is one of the major economies within the Southern African Development Community (SADC) region of Africa. In 2006, the government implemented a comprehensive package of reforms aimed at enhancing service delivery, which included the establishment of a national service delivery strategy, the institutionalisation of a comprehensive monitoring and evaluation system for service delivery, and the development of a framework for public-private partnerships (Helao 2015; Schutte et al. 2020). However, the effectiveness of the various reforms has not materialized in terms of service delivery. The government of Namibia relies on the Town Councils for the provision of services that have a positive

impact on the people. This is because Town Councils are perceived as being closest to the people. As a result, it is where the most basic needs of the people are provided. Therefore, various policy directives are put in place to guide and ensure service delivery by the Town Councils (Schutte et al. 2020). These include the provisions in the Local Authority Act, section 30, and the Regional Council Act, section 28, which emphasised the need for local and regional government to ensure the provision of services to local-regional communities in a sustainable manner, and that services must be provided impartially, fairly, equitably, and without bias (Helao 2015; Kamwanyah et al. 2021). Despite these provisions and other legislative prescriptions regarding service delivery, evidence suggests that Town Councils fall short in the provision of services for the people (Kamwanyah et al. 2021).

Recently, there has been a surge in public outcry regarding the service delivery challenges of Town Councils in Namibia (Kalonda and Govender 2021; Kooper 2021; Sitengu 2021). These challenges (such as water shortages, power outages, and sanitation), have caused a backlog in the process of providing services to residents. While prior studies have mainly focused on the obstacles impeding service delivery in municipalities, little research has been conducted to investigate the influence of leadership on the service delivery of Town Councils in Namibia. Poor leadership and a lack of community involvement have been identified as key factors that impede service delivery. Additional research is needed to investigate the influence of leadership on service delivery in Town Councils in Namibia. This is the justification for this study, which aimed to investigate the role of leadership in accelerating service delivery in times of change.

## 2. The Literature Review

Leadership is defined as the process through which one person motivates a group of people to work towards a common goal (Northouse 2021). This definition of leadership is not the only definition; scholars variously define the concept. Followers constitute an important element in the leadership process, as they legitimise a leader's position (Ford and Harding 2018; Maxwell 2018; Reiley and Jacobs 2016). A leader's role is to set a clear vision, instill confidence, and motivate subordinates towards the accomplishment of collective goals. In Namibia, service delivery by local authorities remains a subject of heated debate and public scrutiny. The process entails the delivery of municipal services to the public. Studies have shown that residents are not satisfied with municipal service delivery in Namibia (Helao 2015; Kalonda and Govender 2021; Kooper 2021). Research on the influence of leadership on service delivery in times of change remains scarce in Namibia. Previous studies have attempted to investigate the challenges hampering service delivery in municipalities (Kolil et al. 2019; Makanyeza et al. 2013; Mbandlwa and Mishi 2020). The literature shows that leadership practices (Kouzes and Posner 2012) can influence service delivery (Caillier 2014; Chen et al. 2015) in Town Councils in various ways. A study found a significant relationship between leadership and service delivery in County Governments (Kolil et al. 2019). The results showed that leadership accounts for a 35% improvement in service delivery.

A significant revelation on how leaders influence service delivery is that the role of a leader is to coordinate activities, improve business processes, motivate employees, secure commitment to the corporate strategy, and align organisational structure with the strategy (Rajasekar 2014). Leadership is placed at the center of the organisation by requiring those in leadership positions to take on multiple roles. Good leaders inspire subordinates to potentially work harder to enhance service delivery and meet objectives as part of an organisation's expectations (Lee and Chuang 2009). Excellent service delivery should be initiated by leadership (Mthembu 2012). The thoughts and feelings to improve teamwork and performance, leading to better customer service, can be influenced through leadership behaviour (Rigii et al. 2019). In other words, effective leadership requires well-coordinated cognitive and behavioural responses. For example, transformational leaders often communicate common goals to their subordinates in clear terms to increase congruence and organisational effectiveness (Northouse 2021). This means that effective

communication and appropriate leadership style(s) can motivate and drive employees' commitment towards the accomplishment of organisational goals or strategic objectives.

Moreover, recent studies support the use of leadership skills to influence followers or subordinates and to enhance service delivery in organisations (Cornelissen and Smith 2022; Fang et al. 2019; Inderjeet and Scheepers 2022). This study is based on the behavioural theory of leadership, which states that a leader's success is determined by his or her behaviour (Northouse 2021). Accordingly, the behavioural approach consists of task and relationship behaviour approaches. Task behaviour involves leaders who engage in spelling out the duties and responsibilities of group members. Comparatively, relationship behaviour entails leaders who make followers feel at ease with one another, with themselves, and with their surroundings.

## 3. Materials and Methods

The purpose of this study was to investigate the role of leadership in accelerating service delivery in a Town Council in Namibia during times of change. A qualitative research approach was employed to gain insight into the participants' experiences and the meanings associated with their perceptions concerning the role of leadership in enhancing service delivery under changing conditions (Bougie and Sekaran 2019; Saunders et al. 2019). To this end, in-depth interviews were conducted with managers and supervisors in the Town Council to assess their understanding of the influence of leadership on service delivery. An exploratory approach was used to gain a more comprehensive understanding of leadership and service delivery in the Town Council.

### 3.1. Research Participants

Since service delivery has been a lingering problem in the public domain, there was a need to critically scrutinise the issues, which necessitated collecting a great deal of data succinctly. To collect meaningful data, the study adopted face-to-face interviews rather than focus groups due to the existence of COVID-19 safety measures. Nonetheless, face-to-face interviews allowed participants to feel more at ease when answering questions than in a focus group. The interviews were conducted at the participants' workplaces, in their offices. This provided them a more relaxing atmosphere to openly express their views in detail without fear or favour. The participants signed a consent form, agreeing to the interviews, which were recorded on tape and thereafter transcribed from audio to text. Before the commencement of interviews, the purpose of the study was reaffirmed to participants, and all ethical considerations such as anonymity, confidentiality, and consent to be interviewed were explained in detail. An interview guide was developed and carefully analysed, after which it was adjusted to capture the relevant information required to meet the objectives of the study. An observation note was also taken during data collection. The order of the questions in the interview guide began with their leadership role(s) in the Town Council, the styles and tactics they use to influence service delivery, the challenges the Town Council faces in service delivery, and the strategies that have been used, or are intended to be used, to mitigate the challenges. The interviews were conducted in Katima Mulilo, the Zambezi Region in Namibia's 14th region. The interviews were held in October 2022. The average duration of the interviews was twenty (20) minutes.

### 3.2. Sample

In this study, the researchers chose samples using a non-probability sampling technique rather than using random selection (Gentles et al. 2015). The selection of data sources from which data were to be obtained to address the study purpose is known as the sampling method (Bougie and Sekaran 2019; Saunders et al. 2019). To collect rich data for qualitative research, the researchers employed a purposive sampling method, otherwise known as purposeful sampling. Hence, to achieve the specific objectives of this study, the researchers employed a purposive sampling technique to select participants who were willing to provide significant responses to the research questions. These participants were

workers who held managerial roles across a number of Town Council departments. The sample size for this study was ten (10). This sample size is suitable for qualitative research (Clarke and Braun 2013). The advantage of purposeful sampling is that it makes it possible for researchers to glean a great deal of knowledge from the data. As a result, the researchers were able to explain how the population will be significantly affected by the findings (Gentles et al. 2015). It is justified by the researchers' desire to comprehend the actual experiences of a comparable group of directors who oversee the provision of services in the Town Council.

### 3.3. Data Analysis

Cross-validation and group discussion are essential to guarantee mutual comprehension, analytical accuracy, and the validity of research outcomes (Kaupa and Atiku 2020; Steinke 2004). To avoid confirmation bias at the analysis stage, the transcribed data were coded independently by the authors; the generated codes were verified by a qualitative researcher within the faculty for accuracy, consistency, and to eliminate any form of subjectivity. The use of an external party was instrumental in analysing the data objectively and codes were revised as suggested by a neutral investigator. At the data analysis stage, the authors discussed the initial themes until an agreement was reached. The results were double-checked by the authors and verified by the neutral investigator to avoid any potential confirmation bias.

NVivo 12 software was employed in this study to conduct the thematic analysis. The use of thematic analysis allows the qualitative data to be organised into themes and subthemes (Dyili et al. 2018; Ganiyu and Genty 2022). Prior to the commencement of the thematic analysis, the data collected were transcribed into text using a Microsoft Word document. Subsequently, the data were imported into the NVivo software in preparation for the analysis. The use of the NVivo software for the thematic analysis allowed for ease of searching for patterns to identify themes. In NVivo, a theme is denoted with a node. A node is similar to a container that houses all patterns relating to a specific theme (Ayandibu et al. 2019; Dyili et al. 2018). Therefore, when a node is opened, all data/patterns relating to the specific theme are revealed. The themes were identified and examined for any trends that appeared in the research results of earlier studies (Bell et al. 2019). Making sense of the data is an active, reflexive process where the researcher's personal experience is crucial. The identification, analysis, and interpretation of qualitative data patterns are also key components of thematic analysis which justify the use of the QSR NVivo 12 software. The participants stressed the influence of leadership on service delivery. The demographic characteristics of the study participants who were purposively selected for data collection are presented in Table 1.

**Table 1.** Demographic characteristics.

| Participants | Gender | Position | Department | Work Experience | Qualifications |
|---|---|---|---|---|---|
| 1 | Male | Supervisor | Road and Store section: Technical Services | 11 years | Diploma in Civil Engineering |
| 2 | Male | Supervisor | Environmental and Public Health | 20 years and above | Diploma in Environmental Health |
| 3 | Male | Supervisor | Water section: Technical Services | 11 years | N3 in Plumbing and Pipe Fitting |
| 4 | Male | Acting Manager | Town Planning section: Town Planning and Land Management | 5 years | Bachelor's degree in civil engineering |

**Table 1.** *Cont.*

| Participants | Gender | Position | Department | Work Experience | Qualifications |
|---|---|---|---|---|---|
| 5 | Female | Supervisor | Procurement Unity: Under Office of the CEO | 5 years and above | MBA |
| 6 | Male | Supervisor | Sewer section: Technical Services | 12 years | Diploma in Plumbing and Pipe Fitting |
| 7 | Female | Manager | Head of Town Planning and Management | 3 years as manager with experience | Bachelor Honors degree |
| 8 | Female | Supervisor | Community Service section: Environmental and Public Health | 5 years | Bachelor Honors in Environmental Health |
| 9 | Male | Manager | Technical Services | 8 years | Bachelor's degree in civil engineering |
| 10 | Male | Manager | Corporate, Bilateral, and Legal Services: Under Office of the CEO | 14 years | Bachelor of Business Administration |

As illustrated in Table 1, the interviews were conducted among the principal employees of the Town Council with adequate knowledge of the research problem and the ability to provide responses to the research questions. Each interview took approximately 20 min per participant. A semistructured interview schedule was employed for the data collection. The use of a semistructured interview allowed the study participants to freely respond to questions (Zulu et al. 2023). The responses provided by each participant resulted in a follow-up question which provided a deeper understanding of the leadership and service delivery issues in the Town Council.

## 4. Results and Discussion

The interviews conducted were aimed at eliciting specific responses regarding the participants perceptions of leadership and service delivery in times of change. Three main themes and nine (9) sub-themes emerged from the thematic analysis conducted using the NVivo 12 software. The main themes that emerged from the thematic analysis included challenges hampering service delivery, leadership roles in service delivery, and strategies for better service delivery. The themes and sub-themes that emerged from the analysed qualitative data are illustrated in Figure 1.

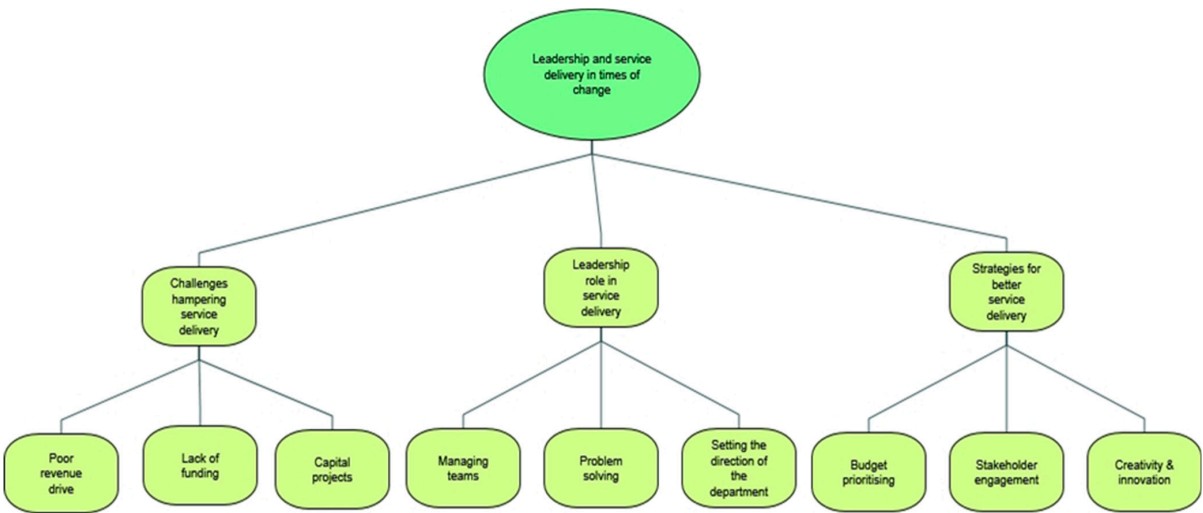

**Figure 1.** Leadership and service delivery.

As illustrated in Figure 1, each of the three main themes birthed three sub-themes each. For emphasis, the sub-themes that emerged from the challenges hampering service delivery, as revealed by the thematic analysis in NVivo, include poor revenue drive, lack of funding, and capital projects. Leadership role in service delivery produced three sub-themes which include managing teams, problem solving, and setting the direction of the department, respectively. Budget prioritising, stakeholder engagement, and creativity and innovation were produced by strategies for better service delivery. The themes and sub-themes are discussed in detail below.

### 4.1. Challenges Hampering Service Delivery

Organisational leadership have the responsibility to constantly develop new business strategies in response to issues that may emerge in a dynamic business environment. The experience in the private sector has resulted in a paradigm shift in the public sector towards more accountability and the adoption of the new style of public service management which aligns with the private sector strategy (Ndevu and Muller 2017; Reddy 2016). This obligation, nevertheless, presents a challenge for public service leaders. The major challenges affecting service delivery, as emerged from the analysed qualitative data, include poor revenue drive of the local council leadership, lack of funding, and capital projects.

### 4.1.1. Poor Revenue Drive

The Town Council relied on various forms of rates and taxes as sources of revenue. The result of the data analysis revealed that the Town Council has difficulty collecting revenue from the residents. This pattern has been present for many years. The amount that the inhabitants are indebted to the local authority is enough to cover the cost of capital projects. The study participants indicated that residents owe more than N\$100 million for municipal services. Participant 4 disclosed, "Town Council had an agreement with a private debt collection agency to obtain all of the funds the residents owed to the Town Council." The study participants suggested that the rates and taxes that the citizens pay for municipal services are what the Town Council use to fund its operations. The views of the two participants below capture the aggregate views of the participants in this study.

> *"Our residents are not fulfilling their responsibilities to pay taxes and charges," They owe the Town Council more than $100 million. This cash could be put to use for various services". (Participant 4)*

> *"In the locality we serve, 80% of the residents struggle to make ends meet on their municipal payments. This is the area where we are meant to make money. The Town Council was told by the government to provide free water to the citizens during COVID-19. NamWater billed us for the water, but the government never reimbursed the Town Council for the water citizens consumed at no cost to them". (Participant 10)*

The participants made it very clear that the Town Council faced financial difficulties despite providing citizens with services. The allegedly owed money by the residents may be enough to address some of its problems. Consistent with this finding, a similar study conducted by Lewis (2017) revealed that government revenue influences service delivery, such as health and infrastructure services. In contrast to the enormous amount of money that the business has currently spent on capital projects, revenue collection might provide the firm with more in the way of cash flow.

### 4.1.2. Lack of Funding

Business operations, the creation of new jobs, and infrastructure development have all been hampered by a lack of funding. The participants emphasised that it is difficult for the Municipal Government to raise enough money to support its operations. Paying service providers is a challenge for the Town Council. A few of these providers have chosen to end their relationships with the Town Council due to unpaid services. According to the participants, the Town Council received funding from the government for capital projects.

Participant 7 affirmed that N$1.8 million was allocated by the government for the financial year 2022–2023. The allocation was not enough to cater to the infrastructural needs of the people. Previous studies have highlighted that the underfunding of local government is a major challenge in service delivery (Kalonda and Govender 2021). The Town Council is now working on capital projects, including building roads and installing sewer systems. However, adequate funding is required for these projects to be completed. The following are some of the participants' opinions concerning the availability of funds:

> *"The Town Council's financial situation is its major challenge. It depends on the availability of finances when you consider what we need to deliver. We lack the funds necessary to adapt to online systems and procedures". (Participant 4)*

> *"Our activities are under pressure due to pending payments from suppliers. Our suppliers stop providing their services to us since we do not pay them on time because there is not money". (Participant 9)*

Participants claimed that the Town Council lacked the resources needed to successfully conduct its functions. The Town Council delayed several projects because of financial limitations. Despite the funding challenges, the residents continued to protest the Town Council authority for the lack of service delivery. The study participants further argue that insufficient funding also creates issues such as departmental equipment and transportation constraints. Participants acknowledged that departments lack the necessary tools to carry out their tasks effectively. The lack of funding is equally affecting the acquisition of equipment for the Technical Services, Environmental, and Public Health departments to carry out their functions effectively.

### 4.1.3. Capital Projects

The Ministry of Urban and Rural Development provides local governments with capital project funding each fiscal year. An essential theme of capital projects was identified by the data gathered from the participants. All the study participants were unanimous that the government underfunded the Town Council. The budgetary allocations for each fiscal year of the one decade were insufficient to carry out its constitutional responsibility with respect to infrastructural development. The view of the participant below captures the sentiments of all the participants in this study.

> *"Town Council does not get government financing. Government funding is only provided to us for capital projects. The construction of water, electricity, and sewage systems will be funded by this money. The Town Council received N$1.8 million from the government for capital projects during the 2022–2023 budget year. The citizens believe that the Town Council is well-funded by the government despite the fact that this money is insufficient to grow the town". (Participant 10)*

### 4.2. Leadership Role in Service Delivery

Leadership involves the ability and a desire to influence the behaviour of others. Team leaders should encourage standards of performance by setting clear expectations of performance and holding team members accountable to them (Katzenbach and Smith 2015). This means that team leaders should create a standard of performance that covers all aspects of the team's work and communicate the standard to their team members. To enhance service delivery in Town Councils, team leaders should monitor the performance of their team members, providing feedback and guidance on how to improve, and rewarding team members who meet or exceed the performance standards (Crews et al. 2019; Sundi 2013; Vito et al. 2014). Furthermore, team leaders should provide the essential resources to support team members to contribute to the success of their teams. For example, Zvavahera (2013) argues that leadership is a complex and multifaceted process which involves the ability to influence followers in a specific direction; goal setting and motivating people through effective communication skills. Similarly, leadership is a behavioural pattern that includes bringing people together to achieve set goals and service delivery (Alex-Nmecha

and David-West 2022). The three sub-themes that emerged from the analysed data in respect to leadership and service delivery were managing teams, problem solving, and setting the direction of the department.

### 4.2.1. Managing Teams

Effective team management has been noted to boost output and fosters a cohesive leadership style within the organisation through team development (Alex-Nmecha and David-West 2022; Zheng et al. 2020). The analysed data revealed that the Town Council employs people from the various Namibian ethnic backgrounds. Managing teams to ensure that they do not feel discriminated against should be a leader's top focus, since today's corporate environment is multicultural (Atiku 2018). The study participants revealed that some issues relating to team management were not properly managed. Given the opinions expressed by Participant 2, some employees, particularly those who felt alienated from the other workers, perceived being discriminated against.

> *"I always show respect to everyone, no matter where they are from. The last time I asked councillors to help me with a problem with some of the senior management here at Town Council, some of them did not treat me with the respect I should have as an employee. A leader should bring people together, not drive them apart". (Participant 2)*

The concern was raised by Participant 2 regarding the selective management style employed by other senior management personnel. Employee discord and hostility result from this, which hinders service delivery and work productivity. Teamwork is impacted because of the inconsistency in the leadership style and the inability to manage diversity among the workforce in the Town Council. The results showed the need to embrace appropriate leadership styles to achieve common goals, which is enhanced service delivery. In this case, leadership styles such as transformational, transactional, and democratic will make a big difference if properly exhibited to influence team members in the Town Council and improve service delivery. Transformational leadership focuses on inspiring and motivating individuals to achieve common goals (Burch and Guarana 2014). It emphasises long-term objectives, encourages creativity and innovation, and prioritises the needs of the collective over those of the individual. Transformational leaders use their influence to motivate and inspire others, while also actively developing the talents and capabilities of their teams (Dong et al. 2017).

Transactional leadership emphasises the use of rewards and punishments to motivate employees (Sundi 2013). This type of leadership focuses on performance and results, and is characterised by setting clear expectations, providing regular feedback, and offering rewards and recognition for the accomplishment of organisational goals. Transactional leaders often use incentives and performance-based pay to achieve short-term objectives (Khan et al. 2014). Participants reported the following as a setback in setting clear expectations and providing regular feedback:

> *"In our department, we communicate information in a specific way. There are individuals who get information, and you must adhere to certain communication routes. But as a leader, you must never undervalue your team members. Sometimes they should provide information". (Participant 4)*

> *"Managers may fail to brief their departments appropriately, which is one of the causes of the oppositional forces. Feedback is crucial since it guides future course of action and throws light on the general operations of the Town Council, as well as what the executive management has to say about the institution's development in terms of service delivery". (participant 10)*

Democratic leadership promotes an atmosphere of participation, collaboration, and consensus-building (Khan et al. 2015). It encourages members of a group to be involved in the decision-making process, allowing for open dialogue, and creating a sense of trust and safety. Democratic leaders strive to create an environment where everyone feels comfortable

to express their ideas and opinions (Iqbal et al. 2015). The following are the participants' opinions on democratic leadership and its influence on service delivery process:

*"The main focus is knowledge exchange. A supervisor does not necessarily have all the answers. We see things differently, but your subordinates may have a solution, therefore you should let them share it". (Participant 3)*

*"I like an environment where everyone is welcome to join and offer their suggestions for how to resolve the problem. This is important since you need other people to help you get outcomes, therefore decisions should take everyone's input into account to make them feel like they are a part of the process". (Participant 5)*

*"I have a policy of having an open door. My employees are always welcome to visit my office to voice their problems and look for solutions. We bring ideas together on how to better our services to the community. We bring ideas together on how to improve our deliverables in our department. I always call them in my office". (Participant 10)*

### 4.2.2. Problem Solving

A highly effective leader possesses the capacity to resolve organisational issues on a regular basis (Cortellazzo et al. 2019; Mbandlwa and Mishi 2020). The participants engage in daily activities that include problem solving. They claimed that the Town Council had been using antiquated procedures and systems to provide services to customers. Land-based delivery and services are among antiquated systems. The participants emphasised as follows:

*"I make sure that all opened job cards are handled as quickly as possible. All complaints brought to the department's attention are handled, and my section informs the manager as necessary". (Participant 6)*

*"It's my responsibility to maintain the Town Council's reputation, I closely interact with the community and deal with their problems, such unrendered services. I talk to people who come to complain about the Town Council's services and work with them to arrive to a mutually agreeable resolution". (Participant 10)*

Modifying system operations could make Town Councils more effective and efficient (Steiss 2019). This could be accomplished by speeding up the method and duration of service delivery to clients. Currently, using the Town Council's services takes a long time. By making services accessible online, clients can use them whenever it is convenient for them without physically visiting the entity. According to the participants, organisations must enhance their systems and procedures to better meet the needs of their clients as the world is constantly changing. The current economic environment requires leaders to rethink their organisational models and develop digital-age strategies (Atiku and Fields 2016). As a result, it is crucial for leaders to improve their problem-solving capabilities to provide sustainable business solutions essential for excellent service delivery.

### 4.2.3. Setting the Direction of the Team

A leader must take the initiative and have faith in the direction of the business to improve performance (Eblie Trudel et al. 2022). Implementing performance-related strategies implies that a leader should have faith in the work they do. Through their actions or personalities at work, leaders can motivate team members to work harder. Employees are more likely to trust and believe in the leader when the leader demonstrates dedication to the course. The study participants affirmed that the workforce mostly rely on the council leadership for direction on service delivery.

*"My main responsibilities are to make sure that the five units in my department are managed successfully. I make sure that all garbage is collected from every home and taken to the dump site. In addition, I oversee the management of the seven independent contractors in charge of waste collection. Additionally, I am in charge of enforcing*

*the Town Council's bylaws and monitoring the town's adequate hygienic standards".
(Participant 2)*

*"I make annual plans for the department's activities based on what the department needs
and supervise the inclusion of new activities". (Participant 3)*

*"My main responsibility is to guarantee that the town is properly planned, including the
establishment of roads, the location of necessary amenities, the inspection of structures,
and the accomplishment of the objectives outlined in the Town Council's strategic plan
for town planning". (Participant 4)*

*"I also make sure that all decisions made by the Town Council are implemented on
schedule. I coordinate all of the Town Council's actions in reference to the decisions
issued by executive management and give recommendations accordingly. Delegating
work to divisions and sub-units, following up when necessary, offering guidance, and
coordinating responses are further responsibilities". (Participant 7)*

However, some head of departments were shown the door due to corruption and
disregard to the constitution. Empirical evidence suggests that corrupt leadership tends to
bring out the worst in a workforce thereby resulting in low motivation at the workplace
(Cornelissen and Smith 2022; Kooper 2021).

### 4.3. Strategies for Better Service Delivery

The entire leadership team in Town Councils has a role to play in influencing their team
members for operational efficiency and the provision of essential services to the populace
(Makanyeza et al. 2013). Additionally, stakeholder engagement is considered a crucial
element in service delivery at the Town Council level. Stakeholder engagement brings
about openness, accountability, and participation of the citizens in governance. The study
participants affirmed that the Town Council entered a public-private partnership (PPP)
agreement with an investment firm to ensure that all local residents could obtain and have
prepaid water meters installed in their homes and offices (Kooper 2021). The PPP agreement
was signed without consultation with the community which resulted in a peaceful protest.
This finding is inconsistent with Masuku and Jili (2019) and Makanyeza et al. (2013) who
affirmed the effectiveness of stakeholder engagement as a viable strategy to enhance service
delivery. However, the three sub-themes emerged from the analysed qualitative data on
the strategies for better service delivery. The sub-themes that are discussed below include
prioritising, stakeholder engagement, and creativity and innovation.

#### 4.3.1. Budget Prioritising

It is evident from the participants that the Town Council lacked the ability to raise
enough money to maintain its operations. Some of the participants considered cutting the
budget and setting priorities for the vital services that the inhabitants required. The quotes
below, from Participant 7 and Participant 10, resonate the views of the majority of the study
participants:

*"Prioritising our needs is necessary. Knowing what is lacking will help us invest
in infrastructure development to draw in investors and turn . . . . . . into a town".
(Participant 7)*

*"In order to meet our goals on a quarterly basis with the limited resources we have, we
must break down our goals into smaller, more manageable goals". (Participant 9)*

*"Management and the procurement department are going to meet to make sure that some
of the less-urgent projects are postponed or suspended until we have enough funding".
(Participant 10)*

Indeed, the participants pointed out that it was critical to consider budget cuts and to
practise appropriate budget control and management at this point in the budgetary crisis
that had enveloped the nation's businesses.

### 4.3.2. Stakeholder Engagement

The opinions of the participants made clear how crucial it was to actively involve important stakeholders in the Town Council's operations. The Roads Authority (RA) and Road Fund Association (RFA) were the main stakeholders highlighted by the participants, as they had been helpful in providing the Town Council with vehicles, traffic equipment, and road development. The participants stated the following:

*"To assist us in managing our sewage systems, we and the University of Namibia (UNAM) have signed a memorandum of understanding. In addition, the Town Council received equipment donations from the RA and RFA for our traffic division as well as vehicles for traffic enforcement. The Town Council had the option of spending millions of dollars on them, but instead, our stakeholders gave them to us. The National Vocational Training Centre will help us train our artisans from the Technical Services Department in the future". (Participant 10)*

### 4.3.3. Creativity and Innovation

The system upgrade and revenue collection models emerged as the sub-themes of creativity and innovation in the data collected from the participants on how to assist the Town Council in overcoming the obstacles to service delivery. System upgrading and digitalization as a new model for revenue collection came out clear from the participants. It was made clear that the Town Council should think about modernising its systems and procedures to enhance service delivery in times of change. Participants complained that the current method of municipal service delivery required too much paperwork. It might be enhanced by taking advantage of electronic service options considering customers' needs in the digital age. The comments from one of our participants are as follows:

*"We are now investigating updating our systems and procedures to better meet customer requests. We want to alter the method we conduct business into online platforms to enable our clients to access our services from anywhere, which would maximise our service delivery process". (Participant 4)*

This result corroborates Fields and Atiku (2016) on the ground that team leaders need to promote collective creativity and eco-innovation for sustainable business solutions and excellent service delivery. Municipalities should evaluate their customers' demands in times of change, which is increasingly driven by technology, and create strategies to address those needs (Bell and Bodie 2012; Joseph and Williams 2022). The reason is that innovation (Kline et al. 2010) is essential in times of change, hence, problem-solving abilities are constantly needed from those in leadership positions (Kotter 2017). System upgrade is considered as an intervention for better service delivery in the Town Council.

One of the factors impacting municipal service delivery in Namibia is revenue collection (Kalonda and Govender 2021). The participants noted that they had a difficult time collecting the rates and taxes owed by the residents. Such revenues served as the Town Council's sole source of funding. To improve the revenue collection, the participants decided to develop a new revenue collection strategy in response to this shortcoming. Participants made the following comments:

*"All people who owe money to the town council will be required to pay their rates and taxes to the entity, Red Force CC, which we have hired to collect money on their behalf". (Participant 4)*

*"We need to consider new income collection strategies, such as introducing new costs like fireman fees. We are not going to get a bailout any time soon from the government". (Participant 10)*

The desire to develop and embark on new modalities to enhance revenue collection in the Town Council was evident. The strategy was similar to the one used by the government when it established the Namibia Revenue Agency (NamRA) to help the Ministry of Finance maximise revenue collection (Söderström and Wangel 2022). Participant 4 disclosed that

the Town Council had an agreement with a private debt collection agency to obtain all the funds that the residents owed to the Town Council. The responsibilities were to be completed by a specific private organisation called Red Force CC on behalf of the Town Council. Furthermore, Participant 10 suggested the need to develop a new model of revenue collection. The participant claimed that the rates and taxes that the citizens ought to pay for municipal services are what the Town Council uses to fund its operations. In light of the fact that these services are not provided for free, they must be paid for. The Town Council will not be able to fulfil its obligations if it is unable to generate income. Residents' failure to pay for water services in the past caused the Town Council to close the water, which was one of the causes of the town's previous water shortages. The supplier may temporarily turn off the water or power if the Town Council does not pay.

### *4.4. Practical Implications*

Leadership practices for effective service delivery in Town Councils is essential in ensuring that all citizens of the town receive the services they need in an effective and efficient manner. Town Councils should have a set of competent leaders to ensure that all services are provided in an equitable and affordable way. This includes ensuring that the Town Council has adequate resources, such as staff and budget, to meet the needs of the community. Leaders should be able to set clear goals and SMART objectives for the Town Council and communicate these goals effectively to the staff and citizens. Leadership in Town Councils should be able to identify and manage potential risks, including financial and operational risks. This includes being able to plan and anticipate problems before they arise. Leaders should be proactive and willing to develop effective strategies to deal with difficult or unexpected situations in times of change.

To provide effective service delivery, Town Councils should be able to respond quickly and effectively to citizen inquiries and complaints. This includes being able to understand citizens needs and provide a timely response in times of change. A Town Council must also be able to keep citizens informed about the services they are providing and be able to address any concerns from the citizens. Service delivery in Town Councils can be enhanced through stakeholders' engagement and involvement in the decision-making process. This includes providing opportunities for citizens to give input into the Town Councils operations, as well as encouraging citizens to participate in Town Council meetings and activities. By engaging citizens (residents) in the decision-making process, Town Councils can ensure that their services are more accurately tailored to the needs of their community. This study provides an opportunity to gain a deeper understanding of leadership and service delivery in Namibia using an inductive approach. The context of the findings provides a more nuanced understanding of the results and bring to light relevant information to enhance service delivery in public service. Therefore, future studies should adopt a quantitative (deductive) approach using inferential statistics to establish the relationship between leadership and service delivery in the public sector, as well as the moderating role of social, political, and economic factors in the interplay.

### 5. Conclusions

The objective of this study was to evaluate the influence of leadership on service delivery in a Town Council in times of crisis. This study holds that leadership styles affect the way services are delivered. Leadership styles such as democratic, transactional, and transformational make this possible. The results suggest that those in leadership positions should exhibit transformational leadership to inspire, mentor, and motivate their subordinates, thus creating a more conducive working environment for improved service delivery. With transformational leadership, subordinates can be motivated to provide quality service to customers. Hence, transformational leadership is instrumental in building workforce creativity and stimulating their levels of motivation to enhance service delivery. Depending on the circumstance, leaders must be conscious of the most appropriate style to influence their stakeholder to achieve common goals. To increase

service delivery, different leadership styles should be exhibited to inspire employees' motivation and performance. This study concludes that managers in Town Councils should take on leadership responsibilities within their departments. The responsibilities include leading a creative workforce, establishing objectives, and creating action plans to achieve the strategic goals. The other responsibilities are managing teams, inspiring employees for excellent service delivery, and attracting investors to improve social infrastructure. The issues affecting service delivery can be resolved through stakeholders' engagement, consultation, community involvement, development of innovative approaches targeted at achieving higher levels of operational efficiency, and the enhancement of service delivery.

**Author Contributions:** Conceptualization, S.O.A. and C.M.K.; methodology, S.O.A., C.M.K. and I.O.G.; software, I.O.G.; validation, S.O.A., C.M.K. and I.O.G.; formal analysis, S.O.A., C.M.K. and I.O.G.; investigation, S.O.A., C.M.K. and I.O.G.; resources, S.O.A., C.M.K. and I.O.G.; data curation, S.O.A., C.M.K. and I.O.G.; writing—original draft preparation, S.O.A., C.M.K. and I.O.G.; writing—review and editing, S.O.A. and C.M.K. visualization, I.O.G.; supervision, S.O.A.; project administration, S.O.A. and C.M.K. All authors have read and agreed to the published version of the manuscript.

**Funding:** The APC was funded by Walter Sisulu University, South Africa.

**Institutional Review Board Statement:** This study was conducted in accordance with the Higher Degree Committee (HDC) of Namibia University of Science and Technology, Namibia.

**Informed Consent Statement:** Informed consent was obtained from all subjects involved in the study.

**Data Availability Statement:** The data generated during and/or analysed during the study are available from the corresponding author on reasonable request.

**Conflicts of Interest:** The authors declare no conflict of interest.

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
