# Peer review of "Leadership and Service Delivery in Times of Change"

_admsci, doi:10.3390/admsci13050125_

Round 1

Reviewer 1 Report

The manuscript explores the challenges and solutions of better public service delivery and identifies the roles of leadership in improving the performance of delivery. 

First of all, the present study uses the inductive approach to identify the type and content of challenge, leadership role, and strategies. While the inductive approach can be useful to generate new ideas, the study should also expend special effort to guard against confirmation bias. The current study confirms the existing literature's findings but fails to provide new insights on the issues of public service delivery.

Second, the materials collected from interviews should be more used to support the discussions of the study. The current MS utilizes the findings of previous studies, not the voices and texts of interviewees.

Third, the contribution and implication of this study is not quietly stated. What is the benefit of confirming the well-known findings with a qualitative, inductive research? 

Reviewer 2 Report

Thank you very much for this opportunity of reviewing your manuscript. I appreciate the authors' effort, as the research scope is relevant to important problems in the real world to be addressed. But the manuscript needs extensive modification before publication. I would say this revision is challenging and hope that you will complete it successfully.

A primary issue in my understanding is that leadership may matter to some extent, but not decisive. Rather than leadership (at the middle manager and supervisor levels), regulations and incentives that are up to political decisions may be more important. If the authors want to emphasize the importance of leadership, they need to prepare more persuasive explanation about the process toward improvement and compare its effect with other factors clearly. In this regard, you should discuss the potential achievements and limitations of appropriate leadership clearly. But in this case, the flow of arguments needs to be revised thoroughly and the paper title may need to be reconsidered as well.

Please refer to my more detailed comments below.

L.138: the typical interview lasted 20 minutes.

Ll.178-179: The interview took approximately 25 minutes…

Not contradictory but a bit confusing.

l.182: "of the and"?

l.200: managing teams (not "themes").

Ll.229-233: This clearly indicates that not the town council level leadership, but the national government level of leadership is necessary.

Ll.284: what structure do you want to make for this sentence?

As mentioned above, I agree that leadership plays a role, but it is much less important than other external factors.

4.2.4 must be 4.3. 4.2.5, 4.2.6 and 4.2.7 must be 4.3.1, 4.3.2, 4.3.3. I am so afraid that the authors were not conscious about the section and sub-section structure. You may feel this is just a careless mistake, but this will not be evaluated like that way.

For some items, practices by the town council and evaluation on them are shown with the informants, but for others without identifying data sources. All the information obtained by the interviews should be mentioned so while if you relied on other sources like the documents compiled by the council, you should tell so explicitly.

Practical implications are likely to be independent from the role of leadership. Of course generally speaking we can tell leadership matters a lot for everything.

Ll.463-464: The argument on appropriate leadership styles of this sort must be shown more in detail earlier. This sentence seems to appear all of a sudden to me.

Overall arguments are not empirical but dependent upon "experts' perception or evaluation". This is acceptable as fact-finding, but you should not insist that you can suggest something recommended by the survey participants.

From time to time, the authors used the terms like business and corporate. Some application of private sector practices are likely to be effective, but just applying these term is a bit confusing.

One more thing about the term, is the council as the study case Town or Municipal Council?

Reviewer 3 Report

Dear Authors,

I have had a chance to revise your manuscript entitled "Leadership and Service Delivery in Times of Change" submitted to Administrative Sciences.

The paper addresses a concern of quality of service in the public sector in the specific case of town councils in Naimbia. More specifically, it investigates the relationship between leadership and service delivery in town councils. I find the underlying problem (service delivery and service quality in public services in the panorama of political interference and poor infrastructure) up-to-date and relevant to a range of audience. However, despite a well done outline of the problem in the section of introduction, I wonder how the investigated relationship (leadership vs service delivery) and applied research method (interviews) could lead to getting a realistic perspective on the problem and, importantly, provide direction for a change. Let me please be explain: the investigated relationship is relevant but how can you isolate the effect of leadership in working (=also social, political, economic, etc.) environment that is, as you refer, characterized by corruption, nepotism, lack of basic resources, poor infrastructure? Wouldn't it be more correct to investigate the ecosystem and account for all these relevant variables (including then the leadership impact)?

Kind regards,

Round 2

Reviewer 1 Report

Many thanks for the authors' efforts to strengthen the manuscript. However, there is a still big issue needs to be explicitly addressed in the paper.

First of all, while this study uses a qualitative research method, the purpose of this study is more deductive. The mismatch or less alignment between research purpose and method should be carefully discussed in the MS.

Second, the effects and importance of leadership needs to be more carefully described and discussed in this study. Again, as this paper uses a confirmatory method to support its argument, the leadership importance and effectiveness should be related with the context of leadership in more details.

Author Response

Many thanks for the authors' efforts to strengthen the manuscript. However, there is a still big issue needs to be explicitly addressed in the paper.

First of all, while this study uses a qualitative research method, the purpose of this study is more deductive. The mismatch or less alignment between research purpose and method should be carefully discussed in the MS.

Response

Thank you very much for you time and the constructive feedback provided to enhance the quality of our submission. The purpose of our study is inductive and qualitative approach was adopted to investigate the role of leadership in accelerating service delivery in a Town Council. Please refer to section 3 (material and methods). The purpose of the study has been fine-tuned for proper alignment (see line113-121).

Second, the effects and importance of leadership needs to be more carefully described and discussed in this study. Again, as this paper uses a confirmatory method to support its argument, the leadership importance and effectiveness should be related with the context of leadership in more details.

Response

Thank you very much for your comments. An exploratory method was adopted for the study. For example, the diagram illustrated in Figure 1 was derived using inductive coding and thematic analysis. The role of leadership in enhancing service delivery in the Town Council was extensively discussed in the manuscript. Please refer to headings 4.2 (leadership role in service delivery), 4.2.1 (managing teams), which established the role team leaders, 4.2.2 (problem solving, which also emphasised the role of leadership in providing solutions to complex business problems), and 4.2.3 (setting the direction of the team). The required leadership importance and effectiveness was provided (see 310-451).

Reviewer 2 Report

My conclusion is accepting your revision in present form, as my comments were generally well reflected in the further revised version.

In fact, after reading the paper again, I had an idea of clearly distinguishing the factors into the two; one type is what the "leaders" in the Town Council can control while the other is what they cannot control. Then particularly under the uncontrollable conditions, you can make an argument on the role of leadership more convincingly. Even though the authors may notice this point well and some parts of the manuscript reflect the idea, but more explicit elaboration is advisable. For example, Figure 1 can be reorganized rather than showing the three main factors just in parallel. By so doing, the role of leadership in the Town Council can be articulated for the unfavorable (and somehow uncontrollable) conditions in a much more persuasive way. Also, the current title can be relevant to the paper contents. I am sure I should not request you to work on this from now, as I should have asked you previously. But please have this idea in your mind and hope to utilize it in your future research activities. 

I also found some other points for further potential improvement such that the too much focus on transformational leadership is not consistent to the descriptions of the three types of leadership in 4.2.1 and you may be able to provide more for information sources beyond the participants. But anyway it is not fair to demand you additionally now. So as mentioned at the beginning, my conclusion is accepting your revision.

Author Response

My conclusion is accepting your revision in present form, as my comments were generally well reflected in the further revised version.

In fact, after reading the paper again, I had an idea of clearly distinguishing the factors into the two; one type is what the "leaders" in the Town Council can control while the other is what they cannot control. Then particularly under the uncontrollable conditions, you can make an argument on the role of leadership more convincingly. Even though the authors may notice this point well and some parts of the manuscript reflect the idea, but more explicit elaboration is advisable. For example, Figure 1 can be reorganized rather than showing the three main factors just in parallel. By so doing, the role of leadership in the Town Council can be articulated for the unfavorable (and somehow uncontrollable) conditions in a much more persuasive way. Also, the current title can be relevant to the paper contents. I am sure I should not request you to work on this from now, as I should have asked you previously. But please have this idea in your mind and hope to utilize it in your future research activities. 

Response

Thank you very much for your time and the constructive feedback provided to enhance the quality of our submission. We will keep the idea in mind for future research activities as suggested. For example, the diagram illustrated in Figure 1 was derived using inductive coding and thematic analysis. The role of leadership in enhancing service delivery in the Town Council was extensively discussed in the manuscript. For example, headings 4.2 (leadership role in service delivery), 4.2.1 (managing teams) which established the role team leaders, 4.2.2 (problem solving, which also emphasised the role of leadership in providing solutions to complex business problems), and 4.2.3 (Setting the direction of the team). 

I also found some other points for further potential improvement such that the too much focus on transformational leadership is not consistent to the descriptions of the three types of leadership in 4.2.1 and you may be able to provide more for information sources beyond the participants. But anyway, it is not fair to demand you additionally now. So as mentioned at the beginning, my conclusion is accepting your revision.

Response

Thank you very much for your open mind and constructive feedback. The required information sources beyond the participants were provided using in-text citations, please refer to lines 353-358. 

Round 3

Reviewer 1 Report

The use of qualitative research method to confirm research arguments is not appropriate while it has some values...

Author Response

The use of qualitative research method to confirm research arguments is not appropriate while it has some values...

Response

Thank you very much for the efforts put into the review process. Please note that no statistical values were presented and/or used to confirm research arguments in the manuscript (admsci-2291646) sent for review.